# Scalable Reporter Assays to Analyze the Regulation of *stx2* Expression in Shiga Toxin-Producing Enteropathogens

**DOI:** 10.3390/toxins13080534

**Published:** 2021-07-29

**Authors:** Martin B. Koeppel, Jana Glaser, Tobias Baumgartner, Stefanie Spriewald, Roman G. Gerlach, Benedikt von Armansperg, John M. Leong, Bärbel Stecher

**Affiliations:** 1Max-von-Pettenkofer Institute, LMU Munich, Pettenkoferstr. 9a, 80336 Munich, Germany; janaglaser89@gmail.com (J.G.); T_Baumgartner@gmx.net (T.B.); sj.spriewald@gmx.de (S.S.); Armansperg@mvp.lmu.de (B.v.A.); 2German Center for Infection Research (DZIF), Partner Site LMU Munich, 80336 Munich, Germany; 3Mikrobiologisches Institut-Klinische Mikrobiologie, Immunologie und Hygiene, Universitätsklinikum Erlangen and Friedrich-Alexander-Universität (FAU) Erlangen-Nürnberg, Wasserturmstraße 3/5, 91054 Erlangen, Germany; Roman.Gerlach@uk-erlangen.de; 4Department of Molecular Biology and Microbiology, Tufts University School of Medicine, 136 Harrison Ave, Boston, MA 02111, USA; John.Leong@tufts.edu

**Keywords:** hemolytic–uremic syndrome, HUS, EHEC, HUSEC, STEC, *E. coli*

## Abstract

Stx2 is the major virulence factor of EHEC and is associated with an increased risk for HUS in infected patients. The conditions influencing its expression in the intestinal tract are largely unknown. For optimal management and treatment of infected patients, the identification of environmental conditions modulating Stx2 levels in the human gut is of central importance. In this study, we established a set of chromosomal *stx2* reporter assays. One system is based on superfolder GFP (sfGFP) using a T7 polymerase/T7 promoter-based amplification loop. This reporter can be used to analyze *stx2* expression at the single-cell level using FACSs and fluorescence microscopy. The other system is based on the cytosolic release of the Gaussia princeps luciferase (*gluc*). This latter reporter proves to be a highly sensitive and scalable reporter assay that can be used to quantify reporter protein in the culture supernatant. We envision that this new set of reporter tools will be highly useful to comprehensively analyze the influence of environmental and host factors, including drugs, small metabolites and the microbiota, on Stx2 release and thereby serve the identification of risk factors and new therapies in Stx-mediated pathologies.

## 1. Introduction

Infections with enterohemorrhagic *E. coli* (EHEC) can cause severe, food-borne disease and pose a significant problem to public health worldwide. The reservoir is mainly cattle, and large-scale EHEC infection outbreaks typically originate from the fecal contamination of vegetables or meat [1,2]. The disease is self-limiting in the majority of cases. Production of the Shiga toxins, Stx1 and Stx2, mediates bloody diarrhea and can cause hemolytic–uremic syndrome (HUS), a life-threatening complication of the infection in a fraction of the infected patients. The rate of EHEC-infected patients affected by HUS ranges between 5 and 30%, which appears to depend on the toxin type and other properties of the EHEC strain. Furthermore, a variety of risk factors for progression to HUS have been identified, including antibiotic treatment, elevated leukocyte count and bloody diarrhea [3,4].

Hitherto, therapy that can effectively prevent the onset of HUS is missing. Moreover, once the patient has progressed to HUS, therapy remains largely symptomatic, including supportive care such as fluid supplementation and plasmapheresis. Antibiotics can be used to treat EHEC infection, but the use of specific antibiotics is contraindicated, as application might stimulate toxin production and/or release [5,6]. A recent outbreak in Germany with 3842 affected patients was caused by a new variant of Stx2 producing *E. coli* (HUSEC O104:H4) [7]. In total, 855 HUS cases with a fatality rate of 4.1% were observed, once again highlighting the need for the development of novel drugs that block the action of Shiga toxins. Indeed, recent progress has been made in the identification of novel drugs, biologicals and vaccination approaches that inactivate Shiga toxin binding and its delivery into the target cell [8,9,10,11,12]. However, to date, evidence for clinical efficacy in patients is lacking. The amount of Stx2 released by EHEC in the gastrointestinal tract is critical for development of the systemic disease. Thus, application of drugs that interfere with *stx* expression or release by the pathogen could prevent Stx2 production in the first place.

Stx2 is an AB_5_ toxin composed of one catalytic subunit A and a pentamer of B subunits, which mediate binding to the Stx receptor globotriaosylceramide (Gb3). Subunit A is translocated into the cell where it eventually leads to a block of protein synthesis and host cell death [13,14]. After release from the pathogen into the gut lumen, Stx2 crosses the epithelial barrier and enters the blood stream via an unknown mechanism where it causes damage to Gb3^+^ renal endothelial cells [15]. Possibly, the intimate attachment of EHEC via attaching and effacing (A/E) lesions to the gut epithelium enhances the chance of Stx2 translocation. Strain HUSEC O104:H4 may attach even more efficiently by its aggregative adherence phenotype [16].

The genes for *stx2AB* are encoded within the late gene region of lambdoid prophages, which are integrated in the *E. coli* chromosome [17]. Stx2AB expression is strictly correlated with induction of the phage lytic cycle by the SOS response [18]. Therefore, DNA damaging agents, including UV radiation, hydrogen peroxide [19] and chemicals (e.g., mitomycin C), can initiate phage DNA replication, Stx2 production and bacterial lysis. Environmental factors that modulate *stx2* expression in vivo are largely unknown. DNA-targeting antibiotics (e.g., quinolones) [20] and others, in particular at sub-inhibitory concentrations, can trigger Stx production [21,22]. Furthermore, quorum sensing, catabolite repression and stress affect *stx2* regulation [23]. In the past, a number of small molecules that interfere with *stx2* expression were characterized. LED209, a small molecule, blocks *stx2* expression via inhibition of QseC-dependent quorum sensing [24]. Nowicki and colleagues showed that isothiocyanates have a repressive effect on prophage induction and Stx2 production [25]. Microbiota-derived metabolites, proteins and small molecules may also play a prominent role in modulating Shiga toxin production [26], representing an invaluable resource to be exploited in the future.

Reporter assays are highly useful to screen for inhibitors of toxin expression. EHEC strains require handling in high-containment biosafety level (BSL)-3** laboratories due to the presence of intact *stx2AB* genes. In most reporter strains, *stx2* is replaced by the respective reporter gene. Therefore, inhibitor screens using *stx2* reporters can be performed under BSL-2 conditions. In the past, various reporter systems have been employed to investigate the regulation of *stx2* expression, including beta-galactosidase [21], alkaline phosphatase [27], photorhabdus luciferase [28], green fluorescent protein (*gfp*) [25] and selectable in vivo expression technology (SIVET) [29,30]. Although these reporter systems have been successfully used to study *stx2* expression, they have certain limitations with respect to scalability, dynamic range and biological readout (e.g., toxin expression and/or release). EHEC strains require handling in high-containment biosafety level (BSL)-3 laboratories due to the presence of intact *stx2AB* genes. In reporter strains, *stx2* is replaced by the respective reporter gene. Therefore, inhibitor screens using *stx2* reporters can be performed under BSL-2 conditions.

In this study, we established two alternative *stx2* reporter assays: one system is based on superfolder GFP (sfGFP) [31], using a T7 polymerase/T7 promoter-based amplification loop. This reporter can be used to analyze *stx2* expression at the single-cell level using FACSs and fluorescence microscopy. The other system is based on the *Gaussia princeps* luciferase (*gluc*) [32]. This highly sensitive and scalable reporter assay can be used to quantify the reporter protein in the culture supernatant and thereby assess the influence of inhibitors on toxin expression and release.

## 2. Results

### 2.1. Generation of a Signal-Amplified sfgfp Reporter to Monitor stx2AB Expression at the Single-Cell Level

In order to analyze regulation of *stx2* expression and prophage induction at the single-cell level, two sets of superfolder GFP (sfGFP) reporter strains were constructed in the background of *E. coli* C600W34 (CW). This strain is lysogenic for the *stx_2a_*-encoding phage 933W, which originates from EHEC O157:H7 but is still classified as BSL-2 due to the absence of other EHEC virulence factors [33]. We chose sfGFP because it exhibits higher resistance to chemical denaturants and improved folding kinetics compared to conventional GFP [31]. In one type of reporter strain (‘lytic reporters’), *stx2A* was exchanged by the sfGFP gene (*sfgfp*; Figure 1A). In this reporter strain (CW^sfgfp^), phage 933W is otherwise intact and can still trigger bacterial lysis upon induction with the DNA-damaging antibiotic mitomycin C (MitC). Furthermore, removal of the catalytically active toxin A subunit is sufficient to downgrade the strain to BSL-2. In the other type, *stx2AB* and four downstream genes, including phage lysis genes (*SR*), were replaced by the reporter gene cassette (Figure 1A). This reporter type does not lyse and is designated ‘non-lytic’ (CW^sfgfpΔlys^).

Next, we characterized the growth behavior of wild-type and both reporter strains in lysogeny broth (LB) and LB containing MitC (0.5 µg/mL). No growth differences were observed in LB (Figure 2A). In contrast, 3h after exposure to MitC, the OD_600_ of lysis-proficient CW^sfgfp^ and reporterless background strain C600W34 (CW) strongly declined, indicating phage lysis, while the 933W-deficient parental strain (C600) and the non-lytic CW^sfgfpΔlys^ continued growing. To study the performance of the reporter strains, we determined the mean fluorescence intensity (MFI) for sfGFP within bacteria by FACS after 5 h (Figure 2B). No signal was observed for both control and reporter strains in LB, showing that the reporter is tightly repressed in the absence of SOS stress. Under MitC-treated conditions, sfGFP MFI was overall slightly increased for all reporter and control strains. This is likely caused by higher the autofluorescence of dead bacterial cells. The lysis-proficient CW^sfgfp^ reporter did not show increased sfGFP MFI compared to C600 or CW, which is likely due to the fact that bacteria inducing the reporter will undergo lysis briefly. Accordingly, for the non-lytic CW^sfgfpΔlys^, a faint but significant increase in sfGFP MFI was observed (*p* < 0.001).

We reasoned that the relatively faint sfGFP signal seen in the non-lytic CW^sfgfp Δlys^ strain is due to the fact that the reporter only carries one copy of the *sfgfp* gene per genome. While this reporter strain could still be valuable for in vitro FACS-based quantification, the signal is too faint for other imaging applications, such as in situ localization of bacteria under infection scenarios. To this end, we used a signal amplification system, based on the T7 polymerase gene T7 pol integrated in the prophage-encoded *stx2* locus, and a T7 promotor–*sfgfp* fusion construct on a medium copy number plasmid (p^PT7sfgfp^) [34]. Based on this system, we generated a lytic (CW^T7pol^ p^PT7sfgfp^) and a non-lytic reporter variant (CW^T7polΔlys^ p^PT7sfgfp^). FACS analysis revealed that the T7 amplification-based reporter system is still tightly repressed in the absence of SOS stress. Moreover, in the absence of the chromosomal T7 polymerase, *sfgfp* was not expressed from p^PT7sfgfp^ (Figure 2B). Upon induction with MitC, the non-lytic CW^T7polΔlys^ p^PT7sfgfp^ yielded higher sfGFP signals compared to that of the single-copy variant CW^sfgfpΔlys^. As expected, only a very low sfGFP signal was seen with lytic CW^T7pol^ p^PT7sfgfp^ due to lysis. Growth behavior was overall very similar to the single-copy variants. Overall, the fluorescence intensity of the non-lytic CW^T7polΔlys^ p^PT7sfgfp^ was judged to be sufficiently high for fluorescence microscopy applications (Figure 2C).

### 2.2. Comparison of Fluc and Gluc luciferase Reporter Strains to Monitor stx2AB Induction and Φstx-Induced Lysis as a Proxy for Stx2 Release

In addition to the sfGFP reporter, we also established a luciferase-based *stx2* reporter system, which offers a higher dynamic signal range and better scalability. The firefly luciferase (Fluc) from the firefly *Photinus pyralis* is one of the most common reporter enzymes employed in high-throughput assays. Fluc generates a bioluminescent signal through oxidation of a luciferin substrate. Furthermore, Gaussia luciferase (Gluc) became recently available, which exhibits higher enzyme stability and strong luminescence activity [32]. Gluc exhibits a high signal-to-noise ratio, and luminescence is linearly proportional to the amount of Gluc protein over five orders of magnitude. In order to analyze the regulation of *stx2AB* expression and prophage induction, two sets of luciferase reporter strains were constructed in the background of *E. coli* C600W34 (CW). Similar to the sfGFP reporters, lytic and non-lytic reporter strains were generated (Figure 1A). For the first, *stx2A* was exchanged by the *fluc* or *gluc* reporter gene cassette (CW^fluc^ and CW^gluc^), and, for the non-lytic variant, *stx2AB* and phage lysis were replaced by the reporter gene cassette (CW^flucΔlys^ and CW^glucΔlys^; Figure 1A).

Next, we characterized growth behavior of wild-type and luciferase reporter strains in LB and LB containing MitC (0.5 µg/mL; Figure 3A) and determined the activity of the two different luciferases (Fluc and Gluc) in the culture supernatant or the bacterial pellet at different time points (Figure 3B,C). Similar to observations made for sfGFP reporters, 3 h after exposure to MitC, the OD_600_ of C600W34 (CW) and lysis-proficient reporter strains strongly declined, indicating phage lysis. No lysis was seen for strains in LB (Figure 3A) or prophage-deficient C600 and lysis-deficient reporter strains (Figure 3A).

In general, luciferase activity in the culture supernatant was significantly higher with the lytic reporter than the non-lytic (Figure 3B), while activity in the pellet fraction was lower (Figure 3C). Overall, Fluc generated lower relative luminescence unit (RLU) levels and a lower dynamic range compared to that of Gluc (Fluc: ~10^3^ vs. Gluc: ~10^6^). Of note, Fluc activity in the supernatant and pellet harvested at later time points (i.e., 7 h and overnight after MitC exposure) was drastically reduced, reflecting its low enzyme stability. In contrast, Gluc activity remained stable once it reached a maximum in the supernatant (CW^gluc^) or the pellet (CW^glucΔlys^). For the lytic reporter CW^gluc^, Gluc activity in the pellet declined 3h after exposure to MitC (Figure 3C), while at the same time, supernatant activity increased (Figure 3B). This reflects the rapid release of the reporter enzyme into the culture supernatant from lysed cells, where it can be readily detected.

### 2.3. The Lytic Gluc Reporter Mirrors Kinetics of Stx Release from E. coli

The lytic reporter strain (CW^gluc^) released Gluc into the culture supernatant within 3 h after MitC induction. Gluc activity in the supernatant reaches a maximum at 5 h, where it remains stable for at least 20 h. Moreover, in the absence of MitC treatment, CW^gluc^ continuously releases low levels of Gluc into the culture supernatant. We hypothesized that Gluc release from CW^gluc^ would reflect the kinetics of phage-mediated Stx2 release from the parental *E. coli* C600W34 strain. In order to test this idea, we cultured CW^gluc^ and the Stx_2a_ producer C600W34 in LB in the presence and absence of MitC (0.5 µg/mL) and quantified Gluc and Stx_2a_ in parallel in the supernatant. The strains exhibit highly similar growth characteristics (Figure 4A). To quantify Stx_2a_ toxin levels, we used a Vero cell assay. Vero cells are highly susceptible to Stx-mediated killing and can therefore be used for detecting and quantifying Shiga toxins [35]. We determined the reciprocal of the supernatant concentration at which >50% of Vero cells were killed (1/(CD50]): The higher 1/(CD50), the more Stx_2a_ is present in the sample. During growth in LB, CW^gluc^ and C600W34 steadily release Gluc (Figure 4B) and Stx_2a_ (Figure 4C), respectively, into the supernatant. This is likely attributable to the low rates of spontaneous prophage induction in the bacterial population. In response to MitC, both Gluc and Stx_2a_ levels are significantly increased 3 h after treatment and reach a maximum at 5 h (~6 log-fold). This determines that the kinetics of Gluc release are indeed comparable to Stx_2a_ release, and the culture supernatant from the lytic reporter CW^gluc^ can be used as a proxy for Stx_2a_ levels in the bacterial culture supernatant. For this reason, but also due to its ease of use, scalability and high sensitivity, the CW^gluc^ could be employed for medium-to-high-throughput applications, such as metabolite, chemical or drug screening.

### 2.4. The Gluc Reporter Assay for Φstx-Induced Lysis Can Be Scaled Up to a 96-Well Format

Next, we aimed to demonstrate that this assay can also be performed in a lower volume format, such as a 96-well plate, e.g., to enable higher throughput screening or automation platforms. We inoculated 250µL of OD_600_ 0.1 (~5 × 10^7^/well) CW^gluc^ or CW^gluc^^Δlys^ in LB (~1 × 10^9^ bacteria/mL) in a 96-well U-bottom plate and incubated at 37 °C with shaking for 20 h. In parallel, the tubes were inoculated with 4 mL of LB harboring the same concentration of reporter strains (~1 × 10^9^ bacteria/mL). Gluc activity was determined in 10 μL of the culture supernatant obtained from the two different formats. Gluc activity from reporter strains grown in 96 wells and tubes was congruent, in particular after 5 h (Figure 5A,B). Therefore, we concluded that the reporter assay can be scaled up to a 96-well format, and the data are comparable to our previous characterization in tube format.

### 2.5. Generation of BSL-2 Gluc Reporter Strains in a *BSL-3** Pathogen*

Despite harboring the *stx_2a_* encoding phage 933W, *E. coli* C600 W34 otherwise does not harbor EHEC virulence factors and is a non-pathogenic derivative of the K12 laboratory strain [33]. We sought to determine if the Gluc reporter system could also be established in a pathogenic, Stx2-producing strain. To this end, we chose the mouse pathogen *Citrobacter rodentium* ϕ*stx*_2dact_ (DBS770), which produces Stx_2dact_ and recapitulates the disease pathology of human EHEC infection in mice [36]. DBS770 is a derivative of *C. rodentium* DBS100 [37]. Due to the presence of *stx*_2dact_, it is considered a BSL-3** level pathogen, and experiments with the strain have to be performed under high containment settings, which impedes experimentation. In the background of DBS770, we generated a lytic (DBS^gluc^; [38]) and non-lytic reporter strain (DBS^gluc^^Δlys^). Notably, the replacement of *stxA*_2dact_ and *stxAB*_2dact_ lysis genes by the *gluc* reporter cassette renders the strains BSL-2. Next, we characterized the reporters in LB with or without exposure to MitC (0.5 µg/mL) in the tube format. Compared to *E. coli* reporters (Figure 3A,B), *C. rodentium* reporters grew slower and reached a lower maximum OD_600_ (Figure 6A). MitC treatment led to a more drastic reduction in growth and earlier lysis (Figure 6B). Since *C. rodentium* DBS100 harbors five intact prophages [39], we assume that these prophages also contribute to lysis. Of note, Gluc activity was detected in the culture supernatant, and this was increased when ϕ*stx*_2dact_ intrinsic lysis genes were intact and the culture stimulated with MitC (Figure 6B).

## 3. Discussion

Stx2 is the major virulence factor of EHEC and is associated with an increased risk for HUS in infected patients. The conditions influencing its expression in the intestinal tract are largely unknown. Reporter gene assays have proven to be important tools for the efficient and high-throughput analysis of factors involved in bacterial virulence gene expression [40,41,42]. In the case of EHEC strains, transcriptional reporters replacing the *stx2* genes have the ancillary benefit that they lead to the downgrading of the pathogen risk class due to deletion of the key virulence factor (ZKBS Az. 6790-10-57, September 2017). Here, we generated a new set of Stx2 reporter strains that can be implemented in different STEC strain backgrounds and used for various applications. The BSL-2 sfGFP reporters allow the monitoring of toxin expression in live bacterial cells or in situ fluorescent microscopic analysis in fixed samples. The BSL-2 Gluc reporter strains are an alternative to classic reporters, as they allow the quantification of toxin release from bacterial cells directly in the culture supernatant. This is a technically simple and robust method that can be readily scaled up to parallelized assay formats. The Gluc reporter will enable testing of the effect of specific drugs, metabolites and peptides on *stx2* expression and toxin release in more detail.

In EHEC strains, the *stx2* genes are encoded on lambdoid prophages, and toxin expression is tightly linked to the production of phages during the phage lytic cycle [43,44]. The genes encoding the Stx2 AB subunits and the *stxAB* promoter element are located downstream of the phage late promoter. Here, they are under control of two subsequent processive antitermination systems, N and Q [45]. The latter one, Q, also promotes the expression of *stxAB* genes by interaction with the RNA polymerase [44]. To minimally interfere with this complex regulatory cascade, we generated a reporter system that allows monitoring of the expression of *stx2* genes in its native genomic context, e.g., on the prophage. We show that a single-copy insertion of the *fluc* or *gluc* reporter genes enables tight regulation with high reporter signal intensity. MitC-mediated triggering of the bacterial SOS response and concomitant induction of the 933W lytic cycle lead to strong induction of both reporters (Figure 3).

Multicopy plasmids have often facilitated gene expression studies to construct reporter strains [46]. Plasmid-based fluorescent protein reporters were generated for both *stx1* and *stx2* expression [47,48] Stx1 reporters were shown to be specifically responsive to cues of the SOS response and iron limitation, the environmental cues for *stx1* expression [49,50]. Using the *stx1-yfp* reporter in combination with a *recA*P*-cfp* reporter for the SOS response, Berger et al. showed that antimicrobial agents inhibiting transcription and translation can prevent Shiga toxin expression, even after induction of the SOS response [47,51]. This confirms the results of others [22,52] and, recently, also a preclinical study in mice [38]. As the use of plasmid-based reporters can lead to copy number artifacts, we reason that the single-copy reporters generated in our work will be a valuable tool, especially for physiological studies. In the case of sfGFP reporters, only the “non-lytic” variants proved useful in detecting sfGFP-positive bacteria, which activated the *stx2* promoter. As *stx2* expression entails phage lysis, in the case of the “lytic” variants, the sfGFP reporter protein is released into the culture supernatant and, therefore, cannot be quantified by FACS (Figure 2). The single-copy insertion of the *sfgfp* leads to a relatively weak fluorescent signal, which is sufficient for FACS-based quantification but too low for fluorescent microscopic applications. In contrast, the integration of a T7 polymerase and plasmid-based signal amplification loop resulted in a bright sfGFP signal while preserving tight regulation.

As bacteria expressing *stx2* are doomed to die by phage lysis, intracellular reporter proteins will inevitably be released in the supernatant where they may not be measurable or rapidly loose activity. The BSL-2 Gluc reporters not only allowed us to monitor toxin gene expression within the bacteria but also released reporter in the culture supernatant. Measuring phage-based reporter release rather than its intrabacterial levels is more relevant for assessing free toxin, which can also induce damage to host cells and be translocated to systemic sites. Thus, the “non-lytic” and “lytic” Gluc reporter variants allow for flexible application, such as quantification of the intrabacterial reporter protein and the reporter protein released by phage lysis in the culture supernatant (Figure 3). We show that the half-life of Gluc activity in the culture supernatant is much higher compared to that of Fluc activity, which can be attributed to the higher enzyme stability of Gluc [53]. Moreover, we show that the reporter system can be used to study toxin release by different pathogenic STEC strains—an important benefit, as subtypes of *stx2* encoding phages are correlated with variable degrees in toxin release [54]. We envision that using our genetic toolset, reporter strains can be constructed in other newly emerging STEC strains to study strain-specific features of *stx2* expression and release.

Shimizu and colleagues generated a chromosome–plasmid hybrid bioluminescent reporter system, using *Photorhabdus luminescens luxCDABE* genes. This reporter system produces both the enzyme (luciferase) and an internal fatty aldehyde substrate and, therefore, enables real-time monitoring of *stx* expression in EHEC [28]. However, the *luxCDABE* system has a much smaller dynamic range compared to that of the Gluc assay and requires the addition of antibiotics to select for the reporter plasmid under some conditions, which may limit its application in high-throughput screening.

In conclusion, the “lytic” Gluc reporter is a scalable reporter system characterized by a high signal/noise ratio, which reports Stx production and/or release. We envision that this new set of reporter tools will be highly useful to screen environmental and host factors, including drugs, small metabolites and the microbiota, on Stx2 release and thereby serve the identification of risk factors and new therapies in EHEC-infected patients.

## 4. Materials and Methods

### 4.1. Generation of Bacterial Mutant Strains and Plasmids 

Bacterial strains and plasmids used in this study are listed in Table 1.

All strains were generated by λ Red recombination. The strains 933W and DBS770 were transformed with pKD46 [60]. For the construction of the 933W strains, plasmids containing the reporter gene (Table 1) flanked by a kanamycin cassette surrounded by Flp-FRT sites (FRT *aphT*. FRT) were used as templates for PCR with the respective primers. For the recombination template, a PCR product was generated, containing the reporter gene and the resistance cassette using primers enclosing homology regions of the target region (~50–60 bp). Due to the identical design of the template plasmids, common reverse primers could be used. The combinations of template plasmid, forward and reverse primers are listed in Appendix A. Correct insertion of the reporter was verified by PCR using the proof primers listed in Appendix A. To construct *C. rodentium* mutants (DBSφ ^gluc^, DBSφ ^gluc^^Δ^^lys^), longer homology regions were required. Gibson assembly was used to create a template plasmid containing the reporter gene with the kanamycin resistance cassette flanked by the 400 bp region homologous to the target region of DBS770. Primers were designed with the NEBuilder tool (https://nebuilder.neb.com/ accessed on 1 June 2021). Genomic DNA of DBS770 was used as a template for the PCRs: GA DBS Stx2 up Fw and GA DBS Stx2 up Rev to amplify the region upstream of *stx2A*, and GA DBS Cm dn Fw and GA DBS Cm dn Rev to amplify the region upstream of the lysis gene and the chloramphenicol cassette that was used to generate DBS770. Thus, in DBSφ ^gluc^^Δ^^lys^, the chloramphenicol cassette was removed. The *gluc* gene and the kanamycin cassette were amplified by PCR with the primers GA DBS GlucKan Fw and GA GlucKan Rev with a linearized pWRG701 as the template. Gibson assembly was performed with the three purified PCR products and pSB377 cut with *Bam*HI and *Not*I according to the manufacturer’s (NEB) instructions. The resulting plasmid pMKB3 was linearized with *Eco*RI and used for λ Red recombination. The correct insertion of the construct was verified by PCR using the primers *stx2A* outside Fw and DBS770 outside Rev.

### 4.2. Bacterial Growth Conditions

If not otherwise stated, *E. coli* and *C. rodentium* strains were grown in 5 mL of LB medium for 12 h under mild aeration at 37 °C in test tubes in a rotor wheel. Antibiotic concentrations used were ampicillin (100 µg/mL) or kanamycin (30 µg/mL). Subcultures were set up (1:100) and grown in a rotor wheel until OD_600_ of ~0.5 was reached. The first subcultures were normalized to an OD_600_ = 0.1 and used for inoculation of experimental cultures in tubes (5 mL, triplicates) or 96-well flat-bottom plates (250 µL/well, triplicates) in LB. Experimental cultures were supplemented with mitomycin C (MitC; Roth; 0.5 μg/mL final concentration) and grown for 4–7 h or overnight (18–22 h) at 37 °C under mild aeration. OD_600_ was recorded at indicated times, and samples were taken for reporter quantification.

### 4.3. Luciferase Assays

At indicated time points, 250 µL of each subculture (in tubes) was taken, directly placed on ice and then spun down for 5min at 14,000× *g*/4 °C. The supernatants were transferred to a 96-well plate. In the case of the 96-well subcultures, at each time point, 50 µL was transferred to a fresh 96-well U-bottom plate placed on ice. This plate was centrifuged for 5 min at 3828× *g*/4 °C. The supernatant was carefully transferred to another plate. All 96-well plates (supernatant) and tubes with the pellets were frozen at −20 °C until the measurement was carried out. The pellets were thawed on ice and resuspended in a 180 µL assay buffer (H_2_O, Tris-HCl 10 mM, NaCl 0.6 M, EDTA 1 mM; adjust to pH 7.8 with HCl). A total of 50 µL of 0.1 mm glass beads (covered with assay buffer, pipetted up and down thoroughly) was added to the samples. Bacteria were lysed in a pre-cooled TissueLyser LT (4 °C) for 5 min at 50 Hz. Afterward, lysed samples were centrifuged for 5 min at 14,000× *g* at 4 °C, and 10 µL of the supernatant was used for measurements. Then, 10 µL supernatant/pellet lysate was transferred into an opaque white 96-well plate for measurement of luminescence levels. Adjacent wells were left empty to avoid spillover of the luminescence of very bright wells into the neighbor wells. For wells with Fluc reporters, luciferin reagent (Tricine 20 mM, (MgCO_3_)_4_Mg(OH)_2_ × 5H_2_O 1 mM, EDTA 0.1 M, D(-) luciferin 470 µM, DTT 33 mM, Li3-coenzym A 270 µM, Mg-ATP 530 µM, glycylglycine 125 µM) and, for Gluc reporters, coelenterazine (CTZ) reagent (coelenterazine 12.5 µM in assay buffer) were used as substrates for the luciferases. Evaluation of the luminescence levels was performed with the CLARIOstar plate reader and a standardized protocol with automated injection and measurement: 40 µL of the substrate (luciferin or CTZ) was injected into the 10 µL of sample followed by 1 s double orbital shaking (4 mm) and a 1 s luminescence measurement (no emission filter; gain 2200; optimized focal height).

### 4.4. Preparation of Samples for Confocal Fluorescence Microscopy 

Bacteria were grown as described above. A total of 250 µL (OD_600_ 1) of culture was spun down for 5 min at 4 °C and 8000 rpm. Pellets were resuspended with 250µL of ice-cold 1× phosphate buffer saline (PBS) and 750 µL of ice-cold 4% paraformaldehyde (PFA) in PBS and incubated for 1 h on ice. Subsequently, bacteria were washed three times with ice-cold PBS. Fixed bacteria were immobilized on poly-L-lysine-coated glass slides (Superfrost Plus, Thermo Scientific, Madrid, Spain) by drying. Immobilized bacteria were additionally fixed for 5min with 4% PFA in PBS and washed 3 times with PBS. Bacterial DNA was stained with 4′6-Diamidin-2-phenylindol (DAPI (1 µg/mL), Roth). Subsequently, bacteria were washed, dried in the dark and mounted with Vectashield (Vector) and then sealed with nail varnish. Using 63× oil objective and a magnification of 1 or 2.4, a minimum of 3 images were taken with a Leica SP5 confocal microscope.

### 4.5. Flow Cytometry

Bacteria were grown as described and diluted in filtered PBS to a concentration of 10^7^ cfu/mL. Data were recorded by a FACS Canto II running FACSDiva software (Aria Becton Dickinson). Data were analyzed using the FlowJo software (Tree Star, Inc., Ashland, OR, USA).

### 4.6. Vero Cell Assay

Supernatants from C600W34 culture experiments were directly sterilized via 0.22 µm in-tube filtration (Corning Spin-X). A 1:10 dilution series of the supernatants were generated across a 96-flat-well plate. Vero cells (Vircell, Granada, Spain, reference number: FTVE, lot number: 11VE151) were grown in T75 flakes containing a cell medium of 5% DMEM (Thermo-Fischer) + Pen-Strep (100 U/mL) + L-glutamine (200 U/mL). At 100% confluency, cells were harvested by trypsinization. A total of 100 µL of media containing 2 × 10^4^ Vero cells was pipetted into each well of the plates with the dilution series of the supernatants. The plates were then incubated for 3 d at 37 °C with 5% CO_2_. The Vero cell assay was evaluated by measuring the optical density after the staining with crystal violet as previously described [64,65]. In short, after the incubation of 3 days, the supernatants in the plates were carefully discarded. A total of 50 µL formalin (2%) was added to each well for 2 min and then flicked out. Next, 50 µL of crystal violet solution (containing 0.13% crystal violet, 5% EtOH and 2% formalin) was added to each well for 2 min and then flicked out. After a washing step with 100 µL of H_2_Odd for 3 min, 150 µL of 50% EtOH was added, and the plates were softly shaken. Finally, absorbance (595 nm) was measured to quantify cells stained by crystal violet. The highest dilution at which at least 50% of Vero cells were killed (CD_50_) was defined by an absorption below 50% of the untreated control.

### 4.7. Statistical Analysis

Statistical analyses were performed with Graph Pad Prism Version 5.01. To compare several groups, ANOVA with Tukey’s multiple comparison test was performed.

## Figures and Tables

**Figure 1 toxins-13-00534-f001:**
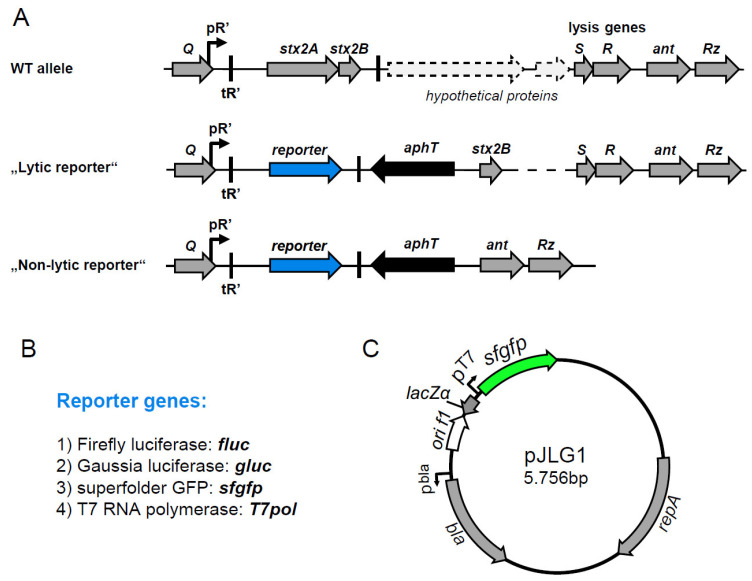
Genetic organization of reporter strains. (**A**) Genetic organization of the *stx*Φ locus wild-type allele and reporter location *E. coli* and *C. rodentium* strains. For the “Lytic reporter”, the reporter gene (list is shown in **B**) and the kanamycin resistance cassette (*aph*T) replaces s*tx2AB* genes, and the phage lysis genes remain intact. For the “Non-lytic reporter”, the reporter gene and *aph*T also replaces part of the phage lysis genes. (**B**) List of reporter genes and abbreviations used in the study. (**C**) Schematic view of the pT7 reporter plasmid pJLG1-harboring T7 promoter-driven *sfgfp* gene.

**Figure 2 toxins-13-00534-f002:**
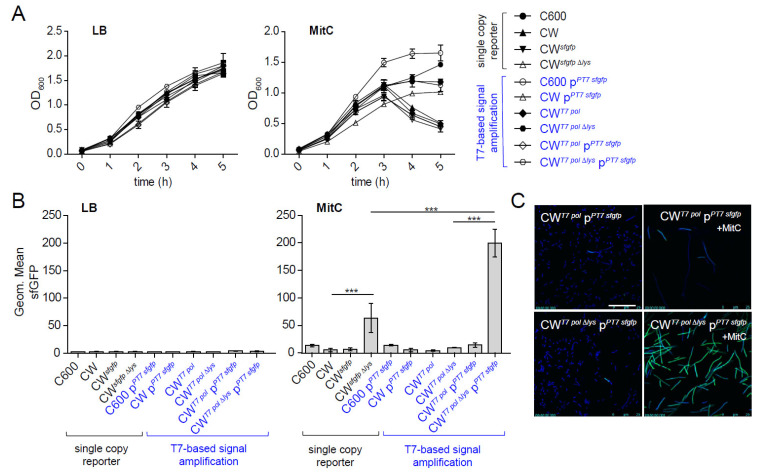
Characterization of *E. coli sfgfp* reporter strains. Single-copy and T7-based signal-amplified *E. coli* reporter strains were grown in LB or in LB supplemented with 0.5 µg/mL MitC at 0 h. (**A**) OD_600_ was recorded at indicated time points (mean ± StD of 3 independent experiments). (**B**) At time point 5 h, bacteria were analyzed for sfGFP signal intensities by flow cytometry. The geometric mean of sfGFP fluorescence (3 independent experiments) is plotted. (**C**) Fluorescent microscopic images of cultures of CW^T7pol^ p^PT7*sfgfp*^ and CW^T7polΔlys^ p^PT7*sfgfp*^ ± MitC. Green: sfGFP. Blue: DAPI. Statistical analysis was performed using ANOVA with Tukey’s multiple comparison test (* *p* < 0.05, ** *p* < 0.01, *** *p* < 0.001).

**Figure 3 toxins-13-00534-f003:**
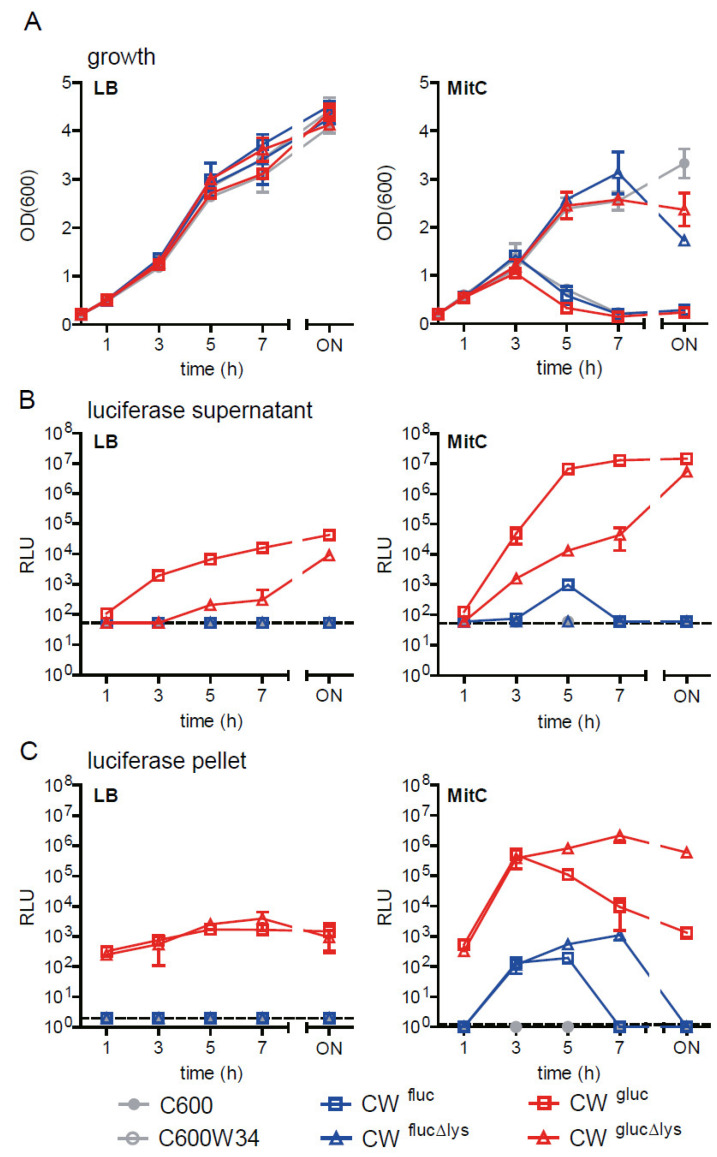
Comparison of *E. coli stx2-*Fluc versus -Gluc reporter strains. Reporter strains (CW^fluc^, CW^flucΔlys^, CW^gluc^ and CW^glucΔlys^) in *E. coli* C600W34 background and control strains were grown in LB (left panels) or in LB supplemented with 0.5 µg/mL MitC (right panels) over the course of 20 h. MitC was added at time point 0. (**A**) Growth kinetics as monitored by OD_600_. Fluc and Gluc luciferase activity in the culture supernatant (**B**) and in intact bacteria (pellet) at indicated time points with respect to MitC treatment (**C**). The relative luminescence units (RLUs) for a fixed volume of 10 µL (mean ± StD of 3 biological replicates) are shown.

**Figure 4 toxins-13-00534-f004:**
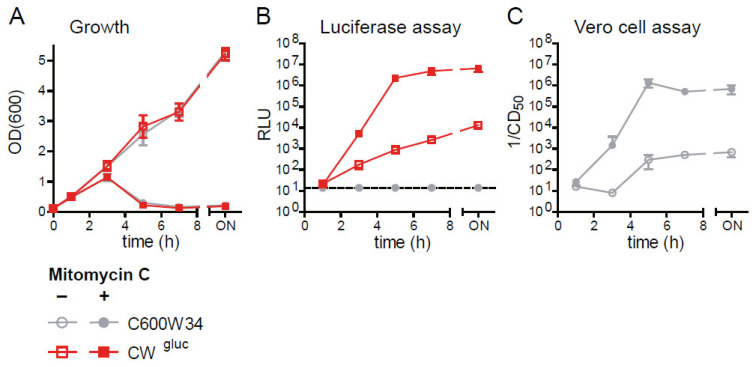
Comparison of Gluc and Stx2 release kinetics in CW^gluc^ and C600W34. (**A**) Overnight cultures of *E. coli* C600W34 and CW^gluc^ were grown in 5 mL of LB medium until an OD_600_ of 0.5. Afterward, bacteria were diluted either in LB or in LB supplemented with MitC (0.5 µg/mL) to an OD_600_ of 0.1. Bacteria were then incubated with shaking at 37 °C for 18 h. At indicated time points, culture supernatant was sampled and analyzed for Gluc activity (**B**) or Stx2 activity by Vero cell assay (**C**). For the Vero cell assay, samples were incubated in 1:2 dilutions (with PBS) in 96-well plates with Vero cells (2 × 10^4^ cells/well) for 3 d. Afterward, cell death was quantified by crystal violet staining. The reciprocal of the highest dilution at which at least 50% of Vero cells were killed 1/(CD50) is depicted. Data are shown as mean ± standard deviation of 3 replicates.

**Figure 5 toxins-13-00534-f005:**
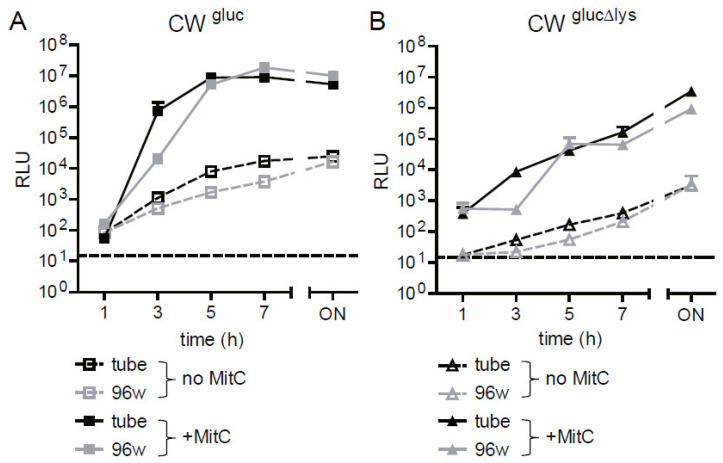
Comparison of Gluc reporter assay in tubes and 96-well format. Precultures of reporter strains CW^gluc^ (**A**) or CW^gluc^^Δlys^ (**B**) in *E. coli* C600W34 background were grown in LB until mid-log phase (OD_600_ of approximately 0.5). Afterward, bacteria were diluted either in LB or in LB supplemented with MitC (0.5 µg/mL) to an OD_600_ of 0.1. The 250 µL/well (96-well plate) and 4 mL/tube were transferred and incubated at 37 °C while shaking. Samples were taken at indicated time points, and Gluc activity was measured as described in Materials and Methods in the culture supernatant depicted as RLU per 10µL supernatant. Data are shown as mean ± standard deviation of 3 replicates.

**Figure 6 toxins-13-00534-f006:**
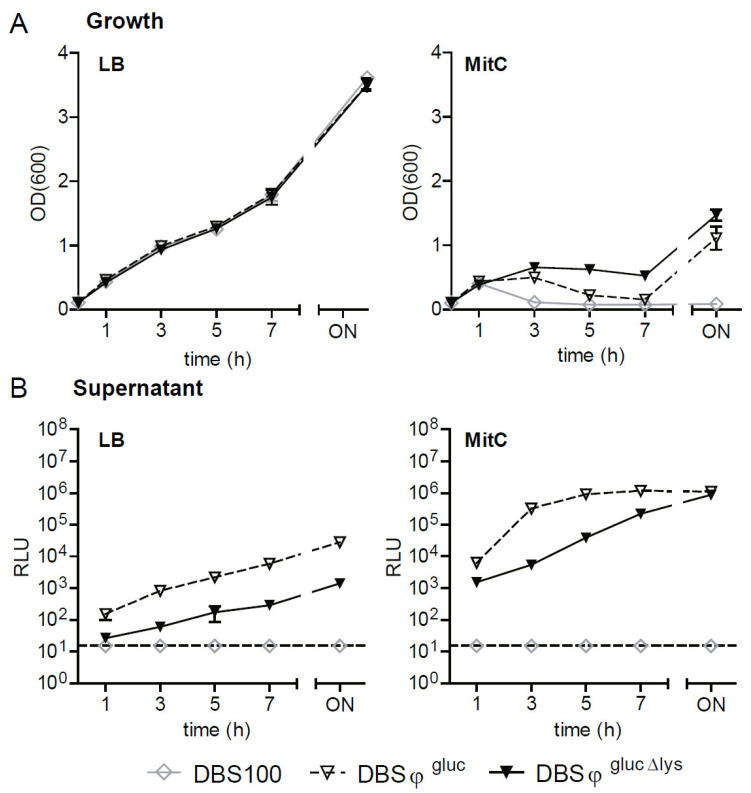
Growth and Gluc release by *C. rodentium* Gluc reporter strains. Precultures of reporter strains DBS^gluc^ or DBS^gluc^^Δ^^lys^ in *C. rodentium* ϕ*stx*_2dact_ background were grown in LB until mid-log phase (OD_600_ of approximately 0.5). Afterward, bacteria were diluted either in LB or in LB supplemented with MitC (0.5 µg/mL) to an OD_600_ of 0.1. Samples were taken at different time points after incubation at 37 °C while shaking, and OD_600_ was determined (**A**,**B**). Gluc activity was measured in the culture supernatant depicted as RLU per 10 µL (**C**,**D**). Data are shown as mean ± standard deviation of 3 replicates.

**Table 1 toxins-13-00534-t001:** Bacteria and plasmids used in this study.

*E. coli* Strains	Designation	Description/Genotype	Reference
DH5α	*Ec* ^DH5^ ^α^	F^−^ Φ80*lac*ZΔM15 Δ(*lac*ZYA-*arg*F) U169 *rec*A1 *end*A1 *hsd*R17(r_k_^−^, m_k_^+^) *pho*A *sup*E44 *thi*-1 *gyr*A96 *rel*A1 λ^−^	Invitrogen
MG1655	*Ec^MG1655^*	*E. coli* K-12 strain MG1655, F^−^ λ^−^ *ilvG*^−^ *rfb*-50 *rph*-1; Rif^R^, Sm^R^	[55,56]
C600	C600		[57]
C600W34	C600W34	*E. coli* C600 transduced with phage 933W from EHEC strain EDL933	[33]
MBK1	CW ^sfgfp^	C600W34 *stx2A::sfgfp aphT*	This study
MBK4	CW ^sfgfp^^Δ^^lys^	C600W34 *stx2AB SR::sfgfp aphT*, phage lysis-deficient strain	This study
MBK6	CW ^fluc^	C600W34 *stx2A::fluc aphT*	This study
MKB7	CW ^fluc^^Δ^^lys^	C600W34 *stx2AB SR::fluc aphT*, phage lysis-deficient strain	This study
JLG5	CW ^gluc^	C600W34 *stx2A::gluc aphT*	[38]
JLG6	CW ^gluc^^Δ^^lys^	C600W34 *stx2AB SR::gluc aphT*, phage lysis-deficient strain	This study
JLG11	CW ^T7pol^	C600W34 *stx2A::T7pol aphT*	This study
JLG12	CW ^T7pol^^Δ^^lys^	C600W34 *stx2AB SR::T7pol aphT*, phage lysis-deficient strain	This study
***C. rodentium* Strains**			
DBS100	DBS100	*Citrobacter rodentium* wild-type strain	[58]
DBS770	DBSφ	*C. rodentium* transduced with phage *stx2*φ17220	[36]
MBK22	DBSφ ^gluc^^Δ^^lys^	DBS770 *stx2AB SR::gluc aphT*, phage lysis-deficient strain	This study
MBK23	DBSφ ^gluc^	DBS770 *stx2A::gluc aphT*	[38]
***Plasmids***			
p3121		High-copy vector, colE1-replicon; carries firefly *luc aphT* flanked by FRT sequences; ampicillin resistance	[59]
pKD46		Temperature-sensitive replication (repA101ts); encodes λ-Red genes (*exo*, *bet*, *gam*); native terminator (tL3) after *exo* gene; arabinose-inducible promoter for expression (P_araB_); encodes *araC* for repression of P_araB_ promoter; ampicillin resistance	[60]
pMBK3		pSB377*stx2AB SR Cm::gluc,aphT*	This study
pMBK4		pSB377*stx2aA::gluc,aphT*	[38]
pSB377		*tetAB oriR6K*	[61]
pJLG1		pM955, P*_T7_*::*sfgfp* Amp^100^	[34]
pJLG2		p2795, *T7 gene 1 aphT* Amp^100^, Kan^30^	[34]
pACYC184			New England Biolabs
pWRG7		High-copy vector, colE1-replicon; carries *sfgfp aphT* flanked by FRT sequences; ampicillin resistance	[46]
pWRG215		High-copy vector, colE1-replicon; carries Gaussia luciferase (flash kinetics) *GlucM43 aphT* flanked by FRT sequences; ampicillin resistance	[62]
pWRG701		High-copy vector, colE1-replicon; carries Gaussia luciferase (glow kinetics) *GlucM43LM110L aphT* flanked by FRT sequences; ampicillin resistance	[38]
pWKS30		Low-copy vector; pSC101-based replicon; ampicillin-resistance marker	[63]

## Data Availability

Data is contained within the article or supplementary material.

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
