# Peer review of "Scalable Reporter Assays to Analyze the Regulation of stx2 Expression in Shiga Toxin-Producing Enteropathogens"

_toxins, 2021, doi:10.3390/toxins13080534_

Round 1

Reviewer 1 Report

See attachment

Author Response

Please see attached word document.

Reviewer 2 Report

In this study, the authors have designed a series of reporter genes/loci for monitoring Stx2 expression by pathogenic or model strains of STEC/EHEC. The work shows that not all solutions are productive. Nevertheless, the authors finally select one reporter system for microscopic observation of toxin production in the bacteria and one system for monitoring toxin release. 

Altogether, this work will be very useful for the study and screening of compounds/factors/probiotics etc. inducing or inhibiting toxin release by pathogenic STEC/EHEC, in vitro and in vivo

The authors indicate that many reporter systems already exist, including beta-Gal, GFP reporters and others. What would be the advantages of the presented tools with respect to the others ? Why initiate the present study ? Hints are given in the Discussion but a few words should justify the work with respect to previous studies, in the introduction.

Fig 1B and C are not clear and only B is described in the figure legend. Does C represent one exemple (sfgfp) of four similar constructs ? Please clarify.

It would be easier to follow the construct strategy if Figure 1 actually mentioned also the names of the WT and modified strains next to the schematized loci. 

The authors should discuss whether or not their systems are limited to the studied strains and if new reporter strains would need to be constructed for instance for each new emerging STEC strain (my understanding).

Author Response

Please see attached word document
